REGISTERED REPORT PROTOCOL

# Protocol for a reproducible experimental survey on biomedical sentence similarity

**Alicia Lara-Clares** *, **Juan J. Lastra-Díaz**, **Ana Garcia-Serrano**

NLP & IR Research Group, E.T.S.I. Informática, Universidad Nacional de Educación a Distancia (UNED), Madrid, Spain

* alara@lsi.uned.es

## Abstract

Measuring semantic similarity between sentences is a significant task in the fields of Natural Language Processing (NLP), Information Retrieval (IR), and biomedical text mining. For this reason, the proposal of sentence similarity methods for the biomedical domain has attracted a lot of attention in recent years. However, most sentence similarity methods and experimental results reported in the biomedical domain cannot be reproduced for multiple reasons as follows: the copying of previous results without confirmation, the lack of source code and data to replicate both methods and experiments, and the lack of a detailed definition of the experimental setup, among others. As a consequence of this reproducibility gap, the state of the problem can be neither elucidated nor new lines of research be soundly set. On the other hand, there are other significant gaps in the literature on biomedical sentence similarity as follows: (1) the evaluation of several unexplored sentence similarity methods which deserve to be studied; (2) the evaluation of an unexplored benchmark on biomedical sentence similarity, called Corpus-Transcriptional-Regulation (CTR); (3) a study on the impact of the pre-processing stage and Named Entity Recognition (NER) tools on the performance of the sentence similarity methods; and finally, (4) the lack of software and data resources for the reproducibility of methods and experiments in this line of research. Identified these open problems, this registered report introduces a detailed experimental setup, together with a categorization of the literature, to develop the largest, updated, and for the first time, reproducible experimental survey on biomedical sentence similarity. Our aforementioned experimental survey will be based on our own software replication and the evaluation of all methods being studied on the same software platform, which will be specially developed for this work, and it will become the first publicly available software library for biomedical sentence similarity. Finally, we will provide a very detailed reproducibility protocol and dataset as supplementary material to allow the exact replication of all our experiments and results.

## Introduction

Measuring semantic similarity between sentences is an important task in the fields of Natural Language Processing (NLP), Information Retrieval (IR), and biomedical text mining, among

**Data Availability Statement:** All relevant data from this study will be made available upon study completion.

**Funding:** ALC UNED predoctoral grant started in April 2019 (BICI N7, November 19th, 2018) https://www.uned.es/ The funders had and will not have a role in study design, data collection and analysis, decision to publish, or preparation of the manuscript.

**Competing interests:** The authors have declared that no competing interests exist.

others. For instance, the estimation of the degree of semantic similarity between sentences is used in text classification [1–3], question answering [4, 5], evidence sentence retrieval to extract biological expression language statements [6, 7], biomedical document labeling [8], biomedical event extraction [9], named entity recognition [10], evidence-based medicine [11, 12], biomedical document clustering [13], prediction of adverse drug reactions [14], entity linking [15], document summarization [16, 17] and sentence-driven search of biomedical literature [18], among other applications. In the question answering task, Sarrouti and El Alaomi [4] build a ranking of plausible answers by computing the similarity scores between each biomedical question and the candidate sentences extracted from a knowledge corpus. Allot et al. [18] introduce a system to retrieve the most similar sentences in the BioC biomedical corpus [19] called Litsense [18], which is based on the comparison of the user query with all sentences in the aforementioned corpus. Likewise, the relevance of the research in this area is endorsed by recent works based on sentence similarity measures, such as the work of Aliguliyev [16] in automatic document summarization, which shows that the performance of these applications depends significantly on the sentence similarity measures used.

The aim of any semantic similarity measure is to estimate the degree of similarity between two textual semantic units as perceived by a human being, such as words, phrases, sentences, short texts, or documents. Unlike sentences from the language in general use whose vocabulary and syntax is limited both in extension and complexity, most sentences in the biomedical domain are comprised of a huge specialized vocabulary made up of all sort of biological and clinical terms, in addition to an uncountable list of acronyms, which are combined in complex lexical and syntactic forms.

Most methods on biomedical sentence similarity are adaptations from methods for the general language domain, which are mainly based on the use of biomedical ontologies, as well as word and sentence embedding models trained on biomedical text corpora. For instance, Socioanglu et al. [20] introduce a set of sentence similarity measures for the biomedical domain, which are based on adaptations from the Li et al. [21] measure. Zhang et al. [22] introduce a set of pre-trained word embedding model called BioWordVec, which is based on a FastText [23] model trained on the titles and abstracts from PubMed articles and term sequences from the Medical Subject Headings (MeSH) thesaurus [24], whilst Chen et al. [25] introduce a set of pre-trained sentence embedding models called BioSentVec, which is based on a Sent2vec [26] model trained on the full text of PubMed articles and Medical Information Mart for Intensive Care (MIMIC-III) clinical notes [27], and Blagec et al. [28] introduce a set of word and sentence embedding models based on the training of FastText [23], Sent2Vec [26], Paragraph vector [29], and Skip-thoughts vectors [30] models on the full-text PubMed Central (PMC) Open Access dataset. Likewise, several contextualized word representation models, also known as language models, have also been adapted to the biomedical domain. For instance, Lee et al. [31] and Peng et al. [32] introduce two language models based on the Bidirectional Encoder Representations from Transformers (BERT) architecture [33], which are called BERT for Biomedical text mining (BioBERT) and Biomedical Language Understanding Evaluation of BERT (BlueBERT), respectively.

Nowadays, there are several works in the literature that experimentally evaluate multiple methods on biomedical sentence similarity. However, they are either theoretical or have a limited scope and cannot be reproduced. For instance, Kalyan et al. [34], Khattak et al. [35], and Alsentzer et al. [36] introduce theoretical surveys on biomedical embeddings with a limited scope. On the other hand, the experimental surveys introduced by Sogancioglu et al. [20], Blagec et al. [28], Peng et al. [32], and Chen et al. [25] among other authors, cannot be reproduced because of the lack of source code and data to replicate both methods and experiments, or the lack of a detailed definition of their experimental setups. Likewise, there are other recent

works whose results need to be confirmed. For instance, Tawfik and Spruit [37] experimentally evaluate a set of pre-trained language models, whilst Chen et al. [38] propose a system to study the impact of a set of similarity measures on a Deep Learning ensembled model, which is based on a Random Forest model [39].

The main aim of this registered report is the introduction of a very detailed experimental setup for the development of the largest and reproducible experimental survey of methods on biomedical sentence similarity with the aim of elucidating the state of the problem, such as will be detailed in the motivation section. Our experiments will be based on our implementation and evaluation of all methods analyzed herein into a common and new software platform based on an extension of the Half-Edge Semantic Measures Library (HESML, http://hesml.lsi. uned.es) [40], called HESML for Semantic Textual Similarity (HESML-STS), as well as their subsequent recording with the Reprozip long-term reproducibility tool [41]. This work is based on our previous experience developing reproducible research in a series of publications in the area, such as the experimental surveys on word similarity introduced in [42–45], whose reproducibility protocols and datasets [46, 47] are detailed and independently confirmed in two reproducible papers [40, 48]. The experiments in this new software platform will evaluate most of the sentence similarity methods for the biomedical domain reported in the literature, as well as a set of unexplored methods which are based on adaptations from the general language domain.

## Main motivations and research questions

Our main motivation is the lack of a reproducible experimental survey on biomedical sentence similarity, which allows the state of the problem to be elucidated in a sound and reproducible way by answering the following research questions:

RQ1.   Which methods get the best results on biomedical sentence similarity?

RQ2.   Is there a statistically significant difference between the best performing methods and the remaining ones?

RQ3.   What is the impact of the biomedical Named Entity Recognition (NER) tools on the performance of the methods on biomedical sentence similarity?

RQ4.   What is the impact of the pre-processing stage on the performance of the methods on biomedical sentence similarity?

RQ5.   What are the main drawbacks and limitations of current methods on biomedical sentence similarity?

Most experimental results reported in this line of research cannot be reproduced for numerous reasons. For instance, Sogancioglu et al. [20] provide neither the pre-trained models used in their experiments nor a detailed guide for replicating them and their software artifacts do not reproduce all of their results. Blagec et al. [28] provide neither a detailed definition of their experimental setup nor their source code and pre-processed data, as well as the pre-trained models used in their experiments. Chen et al. [25] set the state of the art on biomedical sentence similarity by copying results from Blagec et al. [28]; thus, their work allows neither previous results to be confirmed nor are they directly compared with other works. In several cases, biomedical language models based on BERT, such as BioBERT [31] and NCBI-Blue-BERT [32], can be reproduced neither in an unsupervised context nor in any other supervised way, because of the high computational requirements and the non-deterministic nature of the methods used for their training, respectively.

A second motivation is the implementation of a set of unexplored methods which are based on adaptations from other methods proposed for the general language domain. A third motivation is the evaluation in the same software platform of the benchmarks on biomedical sentence similarity reported in the literature as follows: Biomedical Semantic Similarity Estimation System (BIOSSES) [20] and Medical Semantic Textual Similarity (MedSTS) [49] datasets, as well as the evaluation for the first time of the Microbial Transcriptional Regulation (CTR) [50] dataset in a sentence similarity task, despite it having been previously evaluated in other related tasks, such as the curation of gene expressions from scientific publications [51]. A fourth motivation is a study on the impact of the pre-processing stage and NER tools on the performance of the sentence similarity methods, such as that done by Gerlach et al. [52] for stop-words in topic modeling task. And finally, our fifth motivation is the lack of reproducibility software and data resources on this task, which allow an easy replication and confirmation of previous methods, experiments, and results in this line of research, as well as encouraging the development and evaluation of new sentence similarity methods.

### Definition of the problem and contributions

The main research problem tackled in this work is the design and implementation of a large and reproducible experimental survey on sentence similarity measures for the biomedical domain. Our main contributions are as follows: (1) the largest, and for the first time, reproducible experimental survey on biomedical sentence similarity; (2) the first collection of self-contained and reproducible benchmarks on biomedical sentence similarity; (3) the evaluation of a set of previously unexplored methods, as well as the evaluation of a new word embedding model based on FastText and trained on the full-text of articles in the PMC-BioC corpus [19]; (4) the integration for the first time of most sentence similarity methods for the biomedical domain in the same software library called HESML-STS; and finally, (5) a detailed reproducibility protocol together with a collection of software tools and datasets, which will be provided as supplementary material to allow the exact replication of all our experiments and results.

The rest of the paper is structured as follows. First, we introduce a comprehensive and updated categorization of the literature on sentence semantic similarity measures for the general and biomedical language domains. Next, we describe a detailed experimental setup for our experiments on biomedical sentence similarity. Finally, we introduce our conclusions and future work.

## Methods on sentence semantic similarity

This section introduces a comprehensive categorization of the methods on sentence semantic similarity for the general and biomedical language domains, which includes most of the methods reported in the literature. The categorization, shown in Fig 1, is organized into two classes as follows: (a) the methods proposed for the general domain; and (b) the methods proposed for the biomedical domain. For a more detailed presentation of the methods categorized herein, we refer the reader to several surveys on ontology-based semantic similarity measures [43, 45], word embeddings [35, 45], sentence embeddings [34, 53], and neural language models [34, 54].

### Literature review methodology

We conducted our literature review following the next steps: (1) formulation of our research questions; (2) search of relevant publications on biomedical sentence similarity, especially all methods and works whose experimental evaluation is based on the sentence similarity benchmarks considered in our experimental setup; (3) definition of inclusion and exclusion criteria

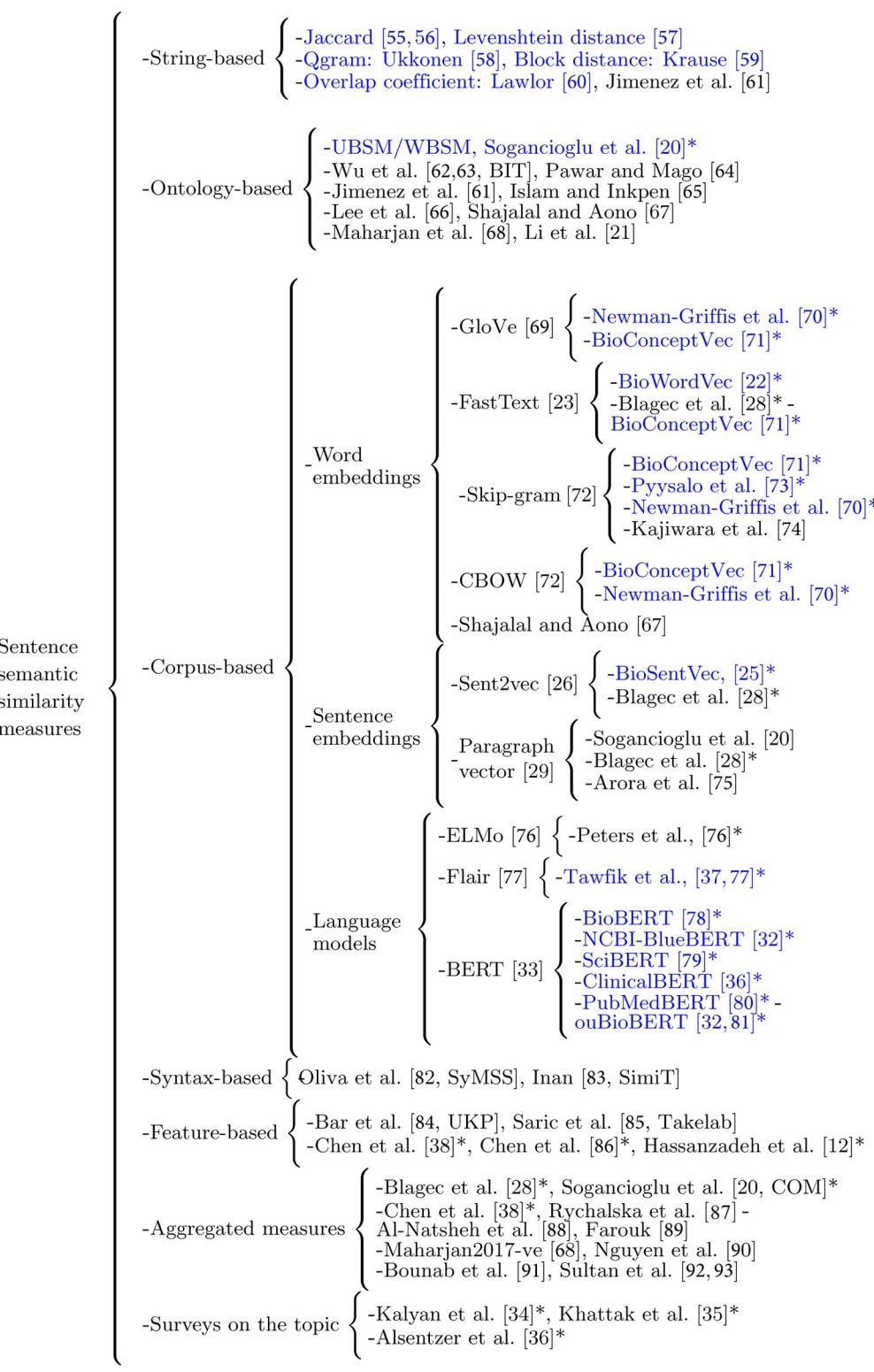

**Fig 1. Categorization of the main sentence similarity methods reported in the literature.** Citations with an asterisk (∗) point out adaptations for the biomedical domain, whilst the citations in blue highlight those methods that will be reproduced and evaluated in our experiments (see Table 8). [12, 20–23, 25, 26, 28, 29, 32–38, 55–93].

of the methods; (4) definition of the study limitations and risks; and (5) definition of the evaluation metrics. Publications on our research topic were mainly searched in the Web Of Science (WOS) and Google Scholar databases, and the SemEval [94–99] and BioCreative/OHNLP [100] conference series. In order to build a first set of relevant works on the topic, we selected a seed set of highlighted publications and datasets on biomedical sentence similarity [20, 21, 25, 28, 31, 49] from the aforementioned information sources. Then, we reviewed all the papers related to sentence similarity which cited any seed publication or dataset. Finally, starting from seed publications and datasets, we extracted those methods that could be implemented and evaluated in our experiments, and we downloaded and checked all the available pre-trained models. Our main goal was trying an independent replication or evaluation of all methods previously evaluated on the biomedical sentence similarity benchmarks considered in our experiments.

## Methods proposed for the general language domain

There is a large corpus of literature on sentence similarity methods for the general language domain as the result of a significant research effort during the last decade. However, the literature for the biomedical domain is much more limited. Research for the general language domain has mainly been boosted by the SemEval Short Text Similarity (STS) evaluation series since 2012 [94–99], which has generated a large number of contributions in the area [84, 85, 92, 101, 102], as well as an STS benchmark dataset [99]. On the other hand, the development of sentence similarity benchmarks for the biomedical domain is much more recent. Currently, there are only three datasets for the evaluation of methods on biomedical sentence similarity, called BIOSSES [20], MedSTS [49], and CTR [50]. BIOSSES was introduced in 2017 and it is limited to 100 sentence pairs with their corresponding similarity scores, whilst MedSTS$_{full}$ is made up by 1,068 scored sentence pairs of the MedSTS dataset [100], which contains 174,629 sentence pairs gathered from a clinical corpus on biomedical sentence similarity. Finally, the CTR dataset includes 171 sentence pairs, but it has not been evaluated yet because of its recent publication in 2019.

Fig 1 shows our categorization of the current sentence semantic similarity measures into six subfamilies as follows. First, string-based measures, whose main feature is the use of the explicit information contained at the character or word level in the sentences to estimate their similarity. Second, ontology-based measures, such as those introduced by Sogancioglu et al. [20], whose main feature is the computation of the similarity between sentences by combining the pairwise similarity scores of their constituent words and concepts [45] based on the Systematized Nomenclature of Medicine Clinical Terms (SNOMED-CT) [103] and WordNet [104] ontologies, and the MeSH thesaurus [24]. Third, corpus-based methods based on the distributional hypothesis [105], such as the work of Pyysalo et al. [73], which states that words sharing semantic relationships tend to occur in similar contexts. The corpus-based methods can be divided into three subcategories as follows: (a) methods based on word embeddings, (b) sentence embeddings, and (c) language models. Methods based on word embeddings combine the word vectors corresponding to the words contained in a sentence to build a sentence vector, such as the averaging Simple Word EMbeddings (SWEM) models introduced by Shen et al. [106], whilst methods based on sentence embeddings directly compute a vector representation for each sentence. Then, the similarity between sentence pairs is calculated using any vector-based similarity metric, such as the cosine function. On the other hand, language models, which explore the concept of Transfer Learning by creating a pre-trained model on a large raw text corpus and fine-tuning those models in downstream tasks, such as sentence semantic similarity, with the pioneering work of Peng et al.

[32]. Fourth, syntax-based methods, which rely on the use of explicit syntax information, as well as the structure of the words that compound the sentences, such as the pioneering work of Oliva et al. [82]. Fifth, feature-based approaches, such as the work of Chen et al. [86], whose main idea is to compute the similarity of two sentences by measuring at different language perspectives the properties that they have in common or not, such as lexical patterns, word semantics and named entities. Finally, aggregated methods, whose main feature is the combination of other sentence similarity methods.

## Methods proposed for the biomedical domain

Like that mentioned in the introduction, most methods on biomedical sentence similarity are adaptations from the general domain, such as the methods which will be evaluated in this work (see Table 8). Sogancioglu et al. [20] proposed a set of ontology-based measures called WordNet-based Similarity Measure (WBSM) and UMLS-based Similarity Measure (UBSM), which are based on the Li et al. [21] measure. All word and sentence embedding models for the biomedical domain in the literature are based on well-known models from the general domain. Pyysalo et al. [73] train a Skip-gram [72] model on document titles and abstracts from the PubMed XML dataset, and all text content of the PMC Open Access dataset. Newman-Griffis et al. [70] and Chen et al. [71] train GloVe [69], Skip-gram, and Continuous Bag of Words (CBOW) [72] models using PubMed information, whilst Zhang et al. [22] and Chen et al. [71] train FastText [23] models using PubMed and MeSH. Blagec et al. [28] introduce a set of neural embedding models based on the training of FastText [23], Sent2Vec [26], Paragraph vector [29], and Skip-thoughts vectors [30] models on the PMC dataset. Chen et al. [25] also introduce a sentence embedding model called BioSentVec, which is based on Sent2vec [26]. Likewise, we also find adaptations from several contextualized word representation models, also known as language models, for the biomedical domain. Tawfik and Spruit [37] evaluate a Flair-based [77] model trained on PubMed abstracts. Ranashinghe et al. [78], Peng et al. [32], Beltagy et al. [79], Alsentzer et al. [36], Gu et al. [80] and Wada et al. [32, 81] introduce BERT-based models [33] trained on biomedical information. However, these later models do not perform well in an unsupervised context because they are trained for downstream tasks using a supervised approach, which has encouraged Ranashinghe et al. [78] to explore a set of unsupervised approximations for evaluating BioBERT [76] and Embeddings for Language Models (ELMo) [76] models in the biomedical domain.

## The reproducible experiments on biomedical sentence similarity

This section introduces a very detailed experimental setup describing our plan to evaluate and compare most of the sentence similarity methods for the biomedical domain. In order to set the state of the art of the problem in a sound and reproducible way, the goals of our experiments are as follows: (1) the evaluation of most of methods on biomedical sentence similarity onto the same software platform; (2) the evaluation of a set of new sentence similarity methods adapted from their definitions for the general-language domain; (3) the setting of the state of the art of the problem in a sound and reproducible way; (4) the replication and independent confirmation of previously reported methods and results; (5) a study on the impact of different pre-processing configurations on the performance of the sentence similarity methods; (6) a study on the impact of different Name Entity Recognition (NER) tools, such as MetaMap [107] and clinic Text Analysis and Knowledge Extraction System (cTAKES) [108], onto the performance of the sentence similarity methods; and finally, (7) a detailed statistical significance analysis of the results.

## Selection of methods

The methodology for the selection of the sentence similarity methods was as follows: (a) identification of all the methods in the biomedical domain that were evaluated in BIOSSES [20] and MedSTS [49] datasets; (b) identification of those methods reported for the general domain not evaluated in the biomedical domain yet; and (c) definition of the criteria for the selection and exclusion of methods.

Our selection criteria for the sentence similarity methods to be reproduced and evaluated herein have been significantly conditioned by the availability of multiple sources of information, as follows: (1) pre-trained models; (2) source code; (3) reproducibility data; (4) detailed descriptions of the methods and experiments; (5) reproducibility guidelines; and finally, (6) the computational requirements for training several models. This work reproduces and evaluates most of the sentence similarity methods for the biomedical domain reported in the literature, as well as other methods that have not been explored in this domain yet. Some of these later unexplored methods are either variants or adaptations of methods previously proposed for the general or biomedical domain, which are evaluated for the first time in this work, such as the WBSM-cosJ&C [20, 43, 109], WBSM-coswJ&C [20, 43, 109], WBSM-Cai [20, 100], UBSM-cosJ&C [20, 43, 109], UBSM-coswJ&C [20, 43, 109], and UBSM-Cai [20, 100] methods detailed in Tables 2 and 3.

**Biomedical methods not evaluated.**   We discard the evaluation of the pre-trained Paragraph vector model introduced by Sogancioglu et al. [20] because it is not provided by the authors, despite this model having achieved the best results in their work. Likewise, we also discard the evaluation of the pre-trained Paragraph vector, sent2vec, and fastText models introduced by Blagec et al. [28], because the authors provide neither their pre-trained models nor their source code and the detailed post-processing configuration used in their experiments. Thus, not all of the aforementioned models can be reproduced.

Tables 1 and 2 detail the configuration of the string-based measures and ontology-based measures that will be evaluated in this work, respectively. Both WBSM and UBSM methods will be evaluated in combination with the following word or concept similarity measures: Rada et al. [111], Jiang&Conrath [112], and three state-of-the-art unexplored measures, called cosJ&C [43], coswJ&C [43], and Cai et al. [110]. The word similarity measure which reports the best results will be used to evaluate the COM method [20]. Table 3 details the sentence similarity methods based on the evaluation of pre-trained character, word, and sentence

**Table 1. Detailed setup for the string-based sentence similarity measures which will be evaluated in this work.** All the string-based measures will follow the implementation of Sogancioglu et al. [20], who use the Simmetrics library [113].

| ID | Method | Detailed setup of each method |
|---|---|---|
| M1 | Qgram [58] | $sim(a,b) = \frac{2 \times |q\text{-}grams(a) \cup q\text{-}grams(b)|}{|q\text{-}grams(a)| + |q\text{-}grams(b)|}$, being $a$ and $b$ sets of q words, and with q = 3. |
| M2 | Jaccard [55, 56] | $sim(a,b) = \frac{|a \cup b|}{|a \cap b|}$, being $a$ and $b$ sets of words of the first and second sentence respectively. |
| M3 | Block distance [59] | $sim(a,b) = 1 - \frac{\sum_{n=1}^{n=|a|+|b|}(v_{an} - v_{bn})}{|a|+|b|}$, being $a$ and $b$ sets of words of the first and second sentence respectively; and $v_a$ and $v_b$ the frequency vectors of $a$ and $b$. |
| M4 | Levenshtein distance [57] | Measures the minimal cost number of insertions, deletions and replacements needed for transforming the first into the second sentence. Insert, delete and substitution cost set to 1. |
| M5 | Overlap coefficient [60] | $sim(a,b) = \frac{|a \cap b|}{|Min(|a|,|b|)|}$, being $a$ and $b$ sets of words of the first and second sentence respectively. |

**Table 2. Detailed setup for the ontology-based sentence similarity measures which will be evaluated in this work.**

| ID | Sentence similarity method | Detailed setup of each method |
|---|---|---|
| M6 | WBSM-Rada [20, 111] | WBSM [20] combined with Rada [111] measure |
| M7 | WBSM-J&C [20, 112] | WBSM [20] combined with J&C [112] measure |
| M8 | WBSM-cosJ&C [20, 43] (this work) | WBSM [20] with cosJ&C [43] measure and Sanchez et al. [109] IC model |
| M9 | WBSM-coswJ&C [20, 43] (this work) | WBSM [20] with coswJ&C [43] measure and Sanchez et al. [109] IC model |
| M10 | WBSM-Cai [20, 110] (this work) | WBSM [20] combined with Cai et al. [110] measure and Cai et al. [110] IC model |
| M11 | UBSM-Rada [20, 111] | UBSM [20] with Rada et al. [111] measure |
| M12 | UBSM-J&C [20, 112] | UBSM [20] combined with J&C [112] measure |
| M13 | UBSM-cosJ&C [20, 43] (this work) | UBSM [20] with cosJ&C [43] measure and Sanchez et al. [109] IC model |
| M14 | UBSM-coswJ&C [20, 43] (this work) | UBSM [20] with coswJ&C [43] measure and Sanchez et al. [109] IC model |
| M15 | UBSM-Cai [20, 110] (this work) | UBSM [20] combined with Cai et al. [110] measure and Cai et al. [110] IC model |
| M16 | COM [20] | $\lambda \cdot$WBSM + $(1 - \lambda) \cdot$ UBSM [20] with $\lambda = 0.5$ and the best word similarity measure |

embedding models that will be evaluated in this work. We will also evaluate for the first time a sentence similarity method, named FastText-SkGr-BioC and detailed in Table 3), which is based on a FastText [23] word embedding model trained on the full text of the PMC-BioC [19] articles. Finally, Table 4 details the pre-trained language models that will be evaluated in our experiments.

**Table 3. Detailed setup for the sentence similarity methods based on pre-trained character, Word Embedding (WE), and Sentence Embedding (SE) models which will be evaluated in this work.**

| ID | Sentence similarity method | Detailed setup of each method |
|---|---|---|
| M17 | Flair [77] | Contextual string embeddings trained on PubMed |
| M18 | Pyysalo et al. [73] | Skip-gram trained on PubMed + PMC |
| M19 | BioConceptVec [71] | Skip-gram WE model trained on PubMed using word2vec program |
| M20 | BioConceptVec [71] | CBOW WE model trained on PubMed using word2vec program |
| M21 | Newman-Griffis et al. [70] | Skip-gram WE model trained on PubMed using word2vec program |
| M22 | Newman-Griffis et al. [70] | CBOW WE model trained on PubMed using word2vec program |
| M23 | Newman-Griffis et al. [70] | GloVe WE model trained on PubMed |
| M24 | BioConceptVec$_{GloVe}$ [71] | GloVe We model trained on PubMed |
| M25 | BioWordVec$_{int}$ [22] | FastText [23] WE model trained on PubMed + MeSH |
| M26 | BioWordVec$_{ext}$ [22] | FastText [23] trained on PubMed + MeSH |
| M27 | BioNLP2016$_{win2}$ [114] | FastText [23] WE model based on skip-gram and trained on PubMed with training setup detailed in [114, table 18] |
| M28 | BioNLP2016$_{win30}$ [114] | FastText [23] WE model based on skip-gram and trained on PubMed with training setup detailed in [114, table 18] |
| M29 | BioConceptVec$_{fastText}$ [71] | FastText [23] WE model trained on PubMed |
| M30 | Universal Sentence Encoder (USE) [115] | USE SE pre-trained model of Cer et al. [115] |
| M31 | BioSentVec [25] | sent2vec [26] SE model trained on PubMed + MIMIC-III |
| M32 | FastText-Skipgram-BioC (this work) | FastText [23] WE model based on Skip-gram and trained on PMC-BioC corpus (05,09,2019) with the following setup: vector dim. = 200, learning rate = 0.05, sampling thres. = 1e-4, and negative examples = 10 |

**Table 4. Detailed setup for the sentence similarity methods based on pre-trained language models which will be evaluated in this work.**

| ID | Sentence similarity method | Detailed setup of each method |
|---|---|---|
| M33 | BioBERT Base 1.0 [31] (+ PubMed) | BERT [33] trained on English Wikipedia + BooksCorpus + PubMed abstracts |
| M34 | BioBERT Base 1.0 [31] (+ PMC) | BERT [33] trained on English Wikipedia + BooksCorpus + PMC full-text articles |
| M35 | BioBERT Base 1.0 [31] (+ PubMed + PMC) | BERT [33] trained on English Wikipedia + BooksCorpus + PubMed abstracts + PMC full-text articles |
| M36 | BioBERT Base 1.1 [31] (+ PubMed) | BERT [33] trained on English Wikipedia + BooksCorpus + PubMed abstracts |
| M37 | BioBERT Large 1.1 [31] (+ PubMed) | BERT [33] trained on English Wikipedia + BooksCorpus + PubMed abstracts |
| M38 | NCBI-BlueBERT Base [32] PubMed | BERT [33] trained on PubMed abstracts |
| M39 | NCBI-BlueBERT Large [32] PubMed | BERT [33] trained on PubMed abstracts |
| M40 | NCBI-BlueBERT Base [32] PubMed + MIMIC-III | BERT [33] trained on PubMed abstracts + MIMIC-III |
| M41 | NCBI-BlueBERT Large [32] PubMed + MIMIC-III | BERT [33] trained on PubMed abstracts + MIMIC-III |
| M42 | SciBERT [79] | BERT [33] trained on PubMed abstracts |
| M43 | ClinicalBERT [116] | BERT [33] trained on PubMed abstracts |
| M44 | PubMedBERT [80] (abstracts) | BERT [33] trained on PubMed abstracts |
| M45 | PubMedBERT [80] (abstracts + full text) | BERT [33] trained on PubMed abstracts + full text |
| M46 | ouBioBERT-Base [81] (Uncased) | BERT [33] trained on PubMed abstracts |

## Selection of language pre-processing methods and tools

The pre-processing stage aims to ensure a fair comparison of the methods that will be evaluated in a single end-to-end pipeline. To achieve this later goal, the pre-processing stage normalizes and decomposes the sentences into a series of components that evaluate the same sequence of words applied to all the methods simultaneously. The selection criteria of the pre-processing components have been conditioned by the following constraints: (a) the pre-processing methods and tools used by state-of-the-art methods; and (b) the availability of resources and software tools.

Most methods receive as input a sequence of words making up the sentence to be evaluated. The process of splitting sentences into words can be carried out by tokenizers for all the methods to be evaluated in this work, such as the well-known general domain Stanford CoreNLP tokenizer [117], which is used by Blagec et al. [28], or the biomedical domain BioCNLPTokenizer [118]. On the other hand, the use of lexicons instead of tokenizers for sentence splitting would be inefficient because of the vast general and biomedical vocabulary. Besides, there would not be possible to provide a fair comparison of the methods because the pre-trained language models have no identical vocabularies.

The tokenized words that conform the sentence, named tokens, are usually pre-processed by removing special characters and lower-casing, and removing the stop words. To analyze all the possible combinations of token pre-processing configurations from the literature, for each method we will replicate the methods used by other authors, such as Blagec et al. [28] and Sogancioglu et al. [20], and we will also evaluate all the pre-processing configurations that have not been evaluated yet. We will also study the impact of pre-processing configurations by not removing special characters nor lower casing and not removing the stop words from the tokens.

Ontology-based sentence similarity methods estimate the similarity of a sentence by exploiting the 'is-a' relations between the concepts in an ontology. Therefore, the evaluation of any ontology-based method in this work will receive a set of concept-annotated pairs of sentences. The aim of the biomedical Named Entity Recognizers (NER) is to identify entities in pieces of raw text, such as diseases or drugs. In this work, we propose to evaluate the impact of three significant biomedical NER tools on the sentence similarity task, as follows: (a) MetaMap [107], (b) cTAKES [108], and (c) MetaMap Lite [119]. MetaMap tool [107] is used by UBSM and COM methods [20] for recognizing Unified Medical Language System (UMLS) [120] concepts in the sentences, which is the standard compendium of biomedical vocabularies. In this work, we will use the default configuration of MetaMap, using all the available semantic types, the MedPost Part-of-speech tagger [121] and with the MetaMap Word-Sense Disambiguation (WSD) module, but restricting UMLS sources to SNOMED-CT and MeSH, which are currently implemented by HESML V1R5 [122]. We will also evaluate cTAKES [108], which has demonstrated to be a robust and reliable tool to recognize biomedical entities [123]. Encouraged by the high computational cost of MetaMap in evaluating large text corpus, Demner-Fushman et al. [119] introduce a lighter MetaMap version, called Metamap Lite, which provides a real-time implementation of the basic MetaMap annotation capabilities without a large degradation of its performance.

## Software integration and contingency plan

To mitigate the impact of potential development risks or unexpected barriers, we have elaborated a contingency plan based on identifying potential risk sources, as well as the testing and integration prototyping of all third-party software components shown in Fig 2. Next, we detail the main risk sources identified in our contingency analysis and the actions carried out to mitigate their impact on our study.

1. *Integration of the biomedical ontologies and thesaurus*. Recently published HESML V1R5 software library [122] integrates the real-time evaluation of ontology-based similarity measures based on MeSH [24] and SNOMED-CT [67], as well as any other biomedical ontology based on the OBO file format [124]. Thus, this risk has been completely mitigated.

2. *External NER tools*. We have confirmed the feasibility of integrating all biomedical NER tools considered in our experiments, such as MetaMap [107] or cTAKES [108], by prototyping the main functions for annotating testing sentences.

3. *Availability of the pre-trained models*. We have already gathered all the pre-trained embeddings [22, 25, 70, 71, 73, 77, 114, 115] and BERT-based language models [31, 32, 79–81, 116] required for our experiments. We have also checked the validity of all pre-trained model files by testing the evaluation of the models using the third-party libraries as detailed below.

4. *Evaluation of the pre-trained models*. The software replication required to evaluate sentence embeddings and language models is extremely complex and out of the scope of this work. For this reason, these models must be evaluated by using the software artifacts used to generate the aforementioned models. Our strategy is to implement Python wrappers for evaluating the available models by using the provided software artifacts as follows: (1) Sent2vec-based models [25] will be evaluated using the Sent2vec library [26]; (2) Flair models [77] will be evaluated using the flairNLP framework [77]; and USE models [115] will be evaluated using the open source platform TensorFlow [125]. All BERT-based pre-trained models will be evaluated using the open-source bert-as-a-service library [126]. On the

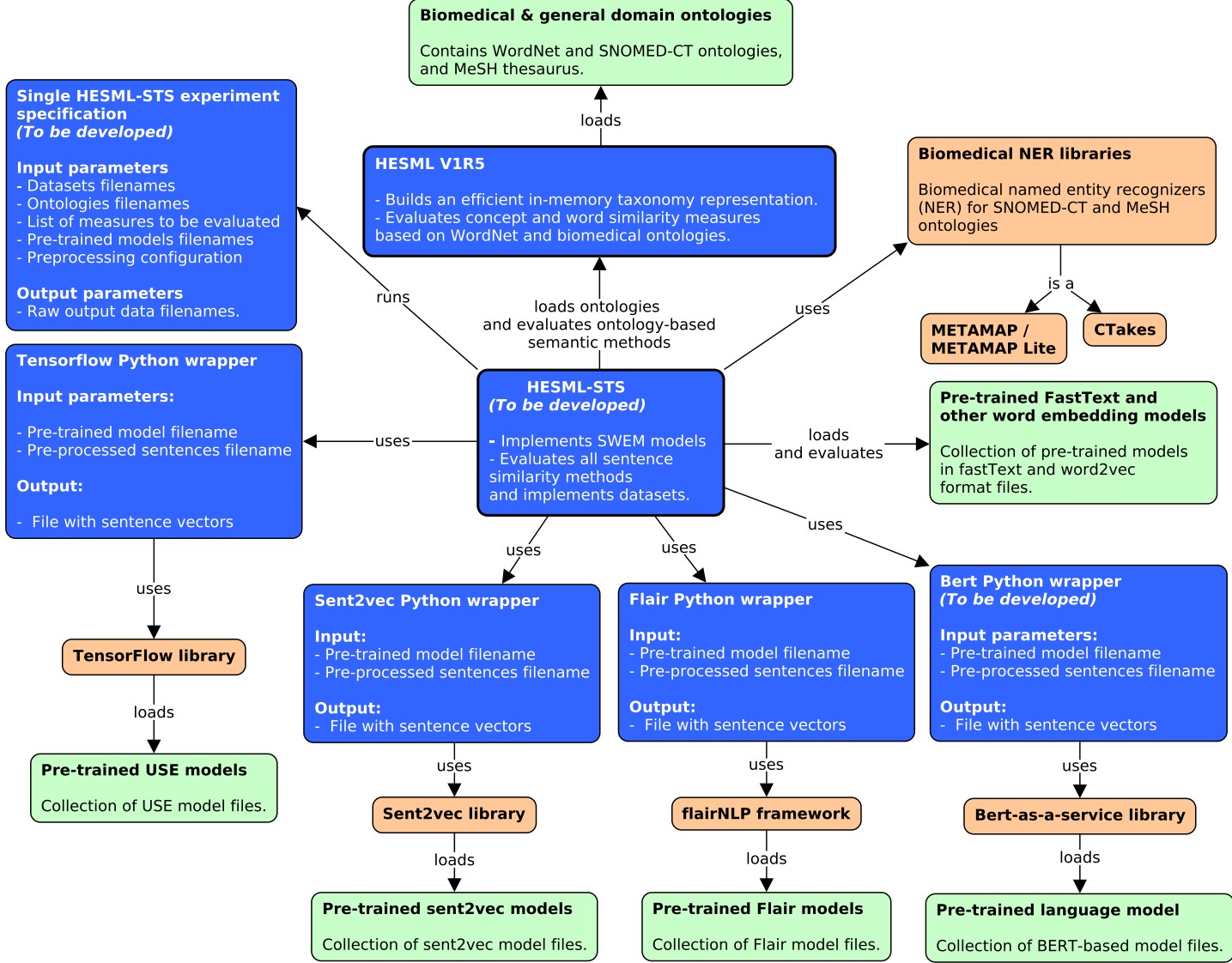

**Fig 2. Concept map detailing the external software components that will be integrated in HESML-STS.** Input data files are shown in green, whilst external software libraries are shown in orange, and software components that will be developed are shown in blue. All experiments will be specified into a single experiment file, which is executed by the HESMLSTSclient program.

other hand, we will develop a parser for efficiently loading and evaluating FastText-based [23] and other word embedding models [22, 70, 71, 73, 114] in the HESML-STS library that will be specially developed for this work. Finally, we have developed all the necessary prototypes to confirm the feasibility of evaluating all the pre-trained models considered in our experiments.

5. *Licensing restrictions.* The licensing restrictions of third-party software components and resources, such as SNOMED-CT [103], MeSH [24] and MetaMap [107], require users to obtain previously a license from the National Library of Medicine (NLM) of the United States to use the UMLS Metathesaurus databases, as well as SNOMED-CT and MeSH. Users will be able to reproduce the experiments of this work by following two alternatives: (1) downloading the third-party software components and integrating them in the

HESML-STS framework as will be detailed in our reproducibility protocol; or (2) by downloading a Docker image file which will contain a pre-installed version of all the necessary software for reproducing our experiments. In the first case, we will publish all the necessary source code, binaries, data, and documentation in Github and Dataverse repositories, to allow the user to integrate restricted third-party software components into the HESML-STS framework. In the second case, users must send a copy of their NLM license to "eciencia@-consorciomadrono.es" to obtain the password to decrypt the Docker file provided as supplementary material.

## Detailed workflow of our experiments

Fig 3 shows the workflow for running the experiments that will be carried out for this work. Given an input dataset, such as BIOSSES [20], MedSTS [49], or CTR [50], the first step is to pre-process all of the sentences, as shown in Fig 4. For each sentence in the dataset (named S1 and S2), the preprocessing phase will be divided into four stages as follows: (1.a) named entity recognition of UMLS [120] concepts, using different state-of-the-art NER tools, such as Meta-Map [107] or cTAKES [108]; (1.b) tokenize the sentence, using well-known tokenizers, such as the Stanford CoreNLP tokenizer [117], BioCNLPTokenizer [118], or WordPieceTokenizer [33] for BERT-based methods; (1.c) lower-case normalization; (1.d) character filtering, which

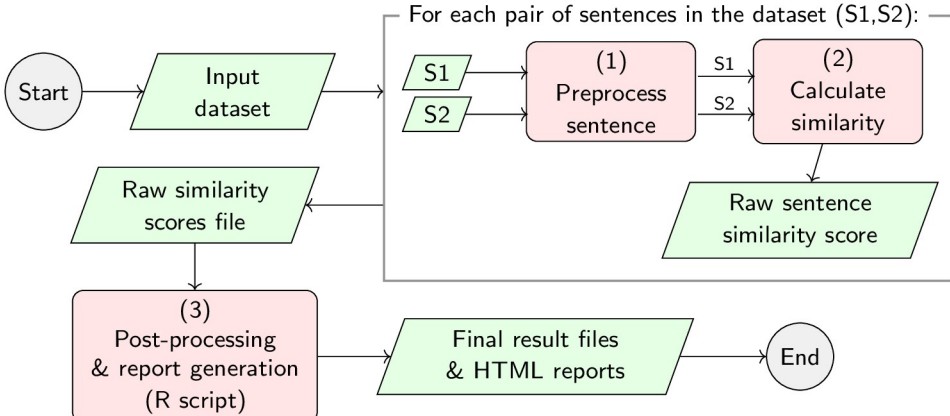

**Fig 3. Detailed experimentation workflow which will be implemented by our experiments to preprocess, calculate the raw similarity scores, and post-process the results contained in the evaluation of the biomedical datasets.** The workflow detailed below produces a collection of raw and processed data files.

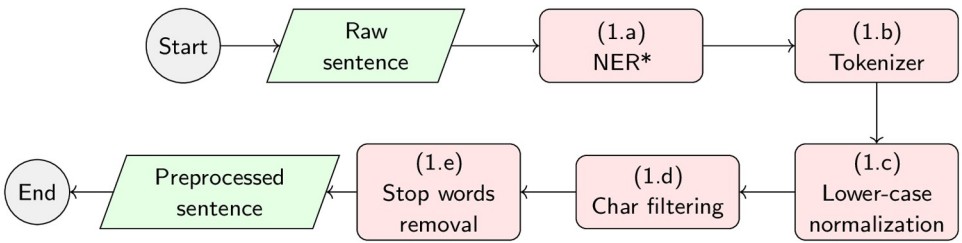

**Fig 4. Detailed sentence preprocessing workflow that will be implemented in our experiments.** The preprocessing stage takes an input sentence and produces a preprocessed sentence as output. (*) The named entity recognizer will be only evaluated in ontology-based methods.

allows the removal of punctuation marks or special characters; and finally, (1.e) the removal of stop-words, following different approximations evaluated by other authors like Blagec et al. [28] or Sogancioglu et al. [20]. Once the dataset is pre-processed in step 1 detailed in Fig 3), the aim of step 2 is to calculate the similarity between each pair of sentences in the dataset to produce a raw output file containing all raw similarity scores, one score per sentence pair. Finally, a R-language script will be used in step 3 to process the raw similarity files and produce the final human-readable tables reporting the Pearson and Spearman correlation values detailed in Table 8, as well as the statistical significance of the results and any other supplementary data table required by our study on the impact of the pre-processing and NER tools.

Finally, we will also evaluate all the pre-processing combinations for each family of methods to study the impact of pre-processing methods on the performance of the sentence similarity methods results, with the only exception of the BERT-based methods. The pre-processing configurations of the BERT-based methods will only be evaluated in combination with the Word-Piece Tokenizer [33] because it is required by the current BERT implementations.

## Evaluation metrics

The evaluation metrics used in this work are the Pearson correlation factor, denoted by $r$ in Eq (1), and the Spearman rank correlation factor, denoted by $\rho$ in Eq (2). The Pearson correlation is invariant regarding any scaling of the data, and it evaluates the linear relationship between two random samples, whilst the Spearman rank correlation is rank-invariant and evaluates the monotonic relationship between two random samples.

$$r = \frac{\sum_{i=1}^{n}(X_i - \overline{X})(Y_i - \overline{Y})}{\sqrt{\sum_{i=1}^{n}(X_i - \overline{X})^2}\sqrt{\sum_{i=1}^{n}(Y_i - \overline{Y})^2}} \tag{1}$$

$$\rho = 1 - \frac{6\sum_{i=1}^{n}d_i^2}{n(n^2-1)}, \qquad d_i = (x_i - y_i) \tag{2}$$

The use of the Pearson correlation to evaluate the task on sentence similarity can be traced back to the pioneering work of Dustin and Alfonsin [127]. On the other hand, both Pearson and Spearman correlation scores have been extensively used to compare the performance of the state-of-the-art methods on biomedical sentence similarity in most works in this line of research [20, 22, 28, 35]. Both aforementioned correlation metrics are also the standard metric for evaluating the task on word similarity [45]. For this reason, we use both aforementioned metrics to evaluate and compare the performance of the methods evaluated herein. However, Spearman's rank correlation has demonstrated to be more reliable in the evaluation of semantic similarity measures of sentences or words in different applications, because it is rank-invariant, and thus, it "provides an evaluation metric that is independent of such data-dependent transformations" [128].

We will use the well-known t-Student test to carry-out a statistical significance analysis of the results in the BIOSSES [20], MedSTS$_{full}$ [49], and CTR [50] datasets. In order to compare the performance of the semantic measures that will be evaluated in our experiments, we use the overall average values of the two aforementioned metrics in all datasets. The statistical significance of the results will be evaluated using the p-values resulting from the t-student test for the mean difference between the values reported by each pair of semantic measures in all datasets, or a subset of them relevant in the context of the discussion. The t-student test is used herein because it is a standard and widely-used hypothesis testing for small and independent data samples with the normal distribution. The p-values are computed using a one-sided t-

student distribution on two paired random sample sets. Our null hypothesis, denoted by $H_0$, is that the difference in the average performance between each pair of compared sentence similarity methods is 0, whilst the alternative hypothesis, denoted by $H_1$, is that their average performance is different. For a 5% level of significance, it means that if the p-value is greater or equal than 0.05, we must accept the null hypothesis. Otherwise, we can reject $H_0$ with an error probability of less than the p-value. In this latter case, we will say that a first sentence similarity method obtains a statistically significantly higher value than the second one in a specific metric or that the former one significantly outperforms the second one.

## Software implementation and development plan

Fig 5 shows a concept map detailing the planned experimental setup to run all experiments planned in this work, as detailed in Table 8. Our experiments will be based on our implementation and evaluation of all methods detailed in Tables 1–4 into a common and new Java software library called HESML-STS, which will be specifically developed for this work. HESML-STS will be based on an extension of the recent HESML V1R5 [122] semantic measures library for the biomedical domain.

All our experiments will be generated by running the *HESMLSTSclient* program shown in Fig 5 with a reproducible XML-based benchmark file, which will generate a raw output file in comma-separated file format (*.csv) for each dataset detailed in Table 5. The raw output files will contain the raw similarity values returned by each sentence similarity method in the evaluation of the degree of similarity between each sentence pair. The final results for the Pearson and Spearman correlation values planned in Table 8 will be automatically generated by running a R-language script file on the collection of raw similarity files using either R or RStudio statistical programs.

Table 6 shows the development plan schedule proposed for this work. We have decomposed the work into seven task groups, called Work Packages (WP), whose deliverables are as follows: (1) Python-based wrappers for the integration of the third-party software components (see Fig 2); (2) HESML-STS library beta 1 version integrated on top of HESML V1R5 (https://github.com/jjlastra/HESML) [122]; (3) HESML-STS beta 1 with an integrated end-to-end pipeline and the XML-based experiment engine; (4) collection of raw output data files generated by running the XML-based reproducible experiments; (5) detailed analysis of the results, including the identification of the main drawbacks and limitations of current methods; (6) reproducible protocol and dataset published in the Spanish Dataverse repository; and finally, (7) submission of the manuscript introducing the study that implements the protocol detailed herein, together with a companion data article introducing our reproducibility protocol and dataset.

## Reproducing our benchmarks

For the sake of reproducibility, we will co-submit a companion data paper with the next work reporting the results of this study, which will introduce a publicly available reproducibility dataset, together with a detailed reproducibility protocol to allow the exact replication of all our experiments and results. Table 7 details the reproducibility software and data that will be published with our next work implementing this registered report. Our benchmarks will be implemented using Java and R languages and could be reproduced in any Java-complaint or Docker-complaint platforms, such as Windows, MacOS, or any Linux-based system. The available software and data will be published on the Spanish Dataverse Network.

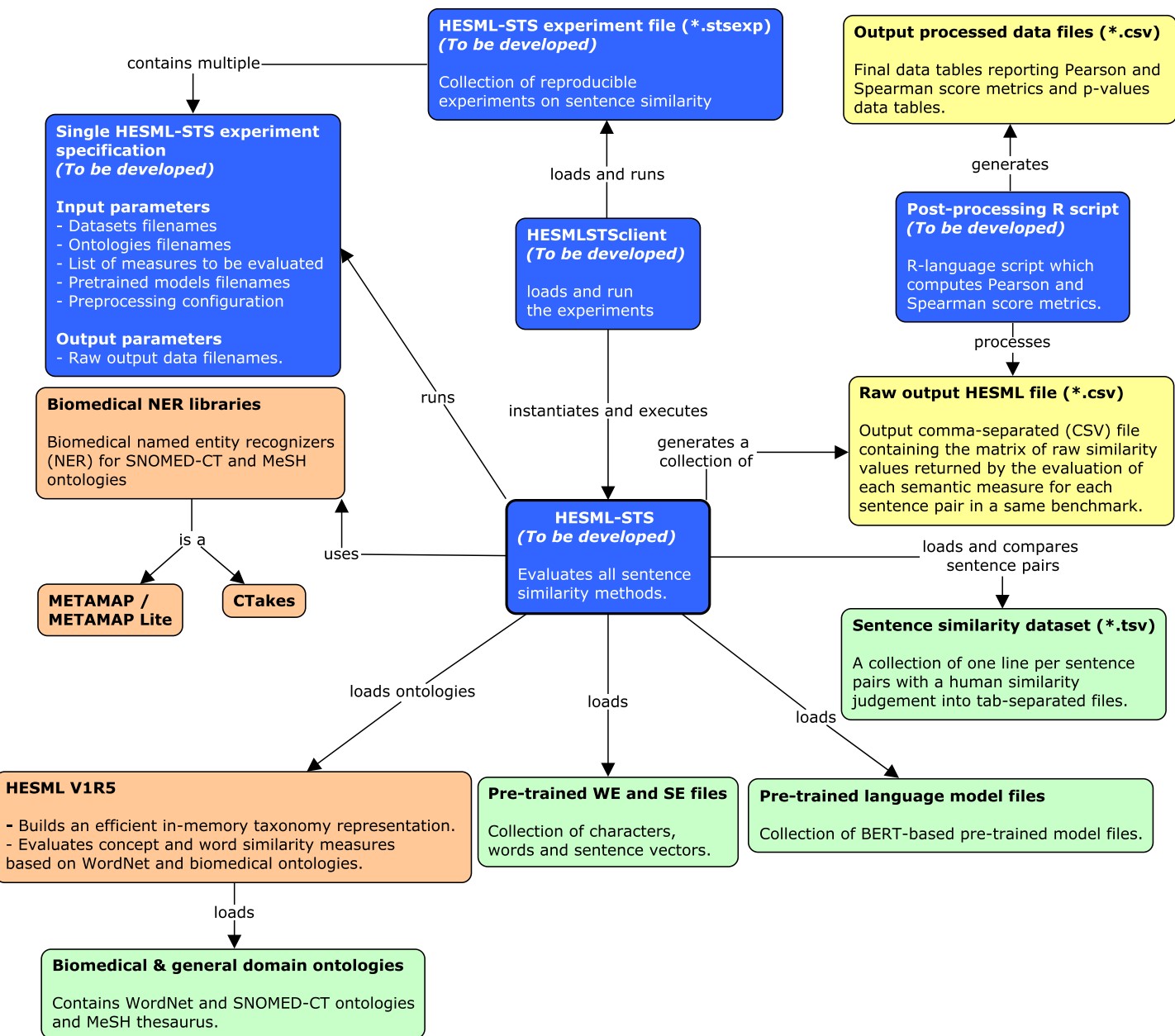

**Fig 5. Concept map detailing the software architecture for our experimental setup.** Input data files are shown in green, whilst output raw and processed data files are shown in yellow, external available software libraries in orange, and software components that will be developed are shown in blue. All experiments will be specified into a single experiment file, which is executed by the HESMLSTSclient program.

**Table 5. Benchmarks on biomedical sentence similarity evaluated in this work.**

| Dataset | #pairs | Corresponding file (*.tsv) in future HESML-STS distribution |
|---|---|---|
| BIOSSES [20] | 100 | BIOSSESNormalized.tsv |
| MedSTS [49] | 1,068 | CTRNormalized_averagedScore.tsv |
| CTR [50] | 170 | MedStsFullNormalized.tsv |

**Table 6. Development plan proposed for this work.**

| Definition of the workpackages and tasks to be developed | Workload (weeks) |
|---|---|
| WP1—Implementation of Python wrappers for third-party components | |
| Task 1.1 Implementation of the BERT Python wrapper | 1 |
| Task 1.2 Implementation of the Sent2vec, Tensorflow, and Flair wrappers | 1 |
| WP2—Software implementation of methods | |
| Task 2.1 Implementation of all pre-processing methods shown in Fig 6 | 2 |
| Task 2.2 Implementation of string-based methods detailed in Table 1 | 1 |
| Task 2.3 Implementation of ontology-based methods detailed in Table 2 | 1 |
| Task 2.4 Implementation of WE and SE methods detailed in Table 3 | 1 |
| Task 2.5 Implementation of BERT-based methods detailed in Table 4 | 1 |
| WP3—Implementation of the automatic reproducible experiments | |
| Task 3.1 Implementation of the benckmark objects and file parsers | 1 |
| Task 3.2 Preparation of the experiment files to evaluate the impact of the pre-processing configurations | 1 |
| Task 3.3 Preparation of the experiment files to evaluate the performance of the methods in the three biomedical sentence similarity datasets | 1 |
| WP4—Evaluation of the entire set of reproducible experiments | |
| Task 4.1 Execution of the pre-processing experiments to generate of all raw output data | 4 |
| Task 4.2 Execution of the method experiments and generation of all raw output data | 2 |
| WP5—Data analysis and results interpretation | |
| Task 5.1 Design and development of the post-processing scripts for the generation of tables and figures | 2 |
| Task 5.2 Data analysis and discussion | 2 |
| Task 5.3 Identification and analysis of the main drawbacks and limitations of current methods | 3 |
| WP6—Design and publication of the reproduciblity protocol and dataset | |
| Task 6.1 Design and validation of the reproducibility dataset | 1 |
| Task 6.2 Design of the reproducibility protocol | 1 |
| Task 6.3 Private publication and validation of the reproducibility dataset | 1 |
| Task 6.4 Software release of the first HESML-STS version | 1 |
| Task 6.5 Creation and validation of the Docker file | 1 |
| Task 6.6 Writing and testing of the reproducibility protocol | 2 |
| Task 6.7 Writing of the companion data article introducing our reproducibility protocol and dataset | 2 |
| WP8—Publishing the results | |
| Task 8.1 Writing and submission of the research article reporting the results of this study and co-submission of the companion data article | 6 |
| Overall estimated workload (weeks) | 39 |

## Detailed results planned

Table 8 shows the methods and datasets that will be evaluated in this work, together with the detailed results which will be generated by our experiments. Finally, any further experimental results resulting from our study on the impact of the pre-processing and NER tools on the performance of the sentence similarity methods will also be reported in our next work, and they could also be reproduced using our aforementioned reproducibility resources.

## Answering our research questions

Next, we explain how our experimental results will allow answering every of our research questions:

**Table 7. Detailed planning of the supplementary reproducibility software and data that will be published with our future work implementing this registered report.**

| Material | Description |
|---|---|
| Reproducibility dataset | Contains all raw input and output data files, pre-trained model files, and a long-term reproducibility image based on ReproZip or Docker, which will be publicly available in the Spanish Dataverse Network. |
| Companion data article | Data and methods article introducing our reproducibility protocol and dataset to allow the independent replication of our experiments and results. |
| HESML-STS software library | Release of the new HESML-STS library. This library will be integrated into a forthcoming HESML version published both in Github and the Spanish Dataverse Network under CC By-NC-SA-4.0 license. |
| HESML-STS software paper | Software article introducing our sentence similarity library, called HESML-STS, which will be especially developed for this work. |

RQ1. Table 8 will report the Pearson and the Spearman rank correlation factors in the evaluation of the three datasets. Therefore, we will draw up our conclusions by comparing the performance of both metrics. However, we will set the best overall performing methods using the Spearman correlation results because of its better predictive nature in most extrinsic tasks, as pointed out in section "Evaluation Metrics".

RQ2. We will use a t-Student test between the Spearman correlation values obtained by each pair of methods in the evaluation of the three proposed datasets as a means to set the statistical significance of the results. Thus, we will say that a method significantly outperforms another one resulting p-values are less or equal than 0.05. The t-Student test will be based on the Spearman rank correlation value for the same reasons detailed above.

RQ3. Table 9 details the methods and biomedical NER tools that will be evaluated in this work. We will consider only ontology-based methods since word and sentence pre-trained models have been trained on raw texts and do not contain UMLS concepts. To make a fair comparison of the methods, we will evaluate them using the best pre-processing configuration defined by a selection of the tokenizer, lower-case normalization, char filtering, and stop words list. Our analysis and discussion of the results will be based on comparing the Pearson and Spearman correlation values reported for each method. However, we will set the best overall performing NER tool using the Spearman rank correlation results like the remaining research questions.

RQ4. Fig 6 details all the possible combinations of pre-processing configurations that will be evaluated in this work. String, word and sentence embedding, and ontology-based methods, will be evaluated using all the available configurations except the WordPiece-Tokenizer [33], which is specific to BERT-based methods. Thus, BERT-based methods will be evaluated using different char filtering, lower casing normalization, and stop words removal configurations. We will use the Pearson and Spearman's correlation values to determine the impact of the different pre-processing configurations on the evaluation results. However, we will set the best overall performing pre-processing configuration using the Spearman rank correlation results like the remaining research questions.

RQ5. Our methodology for identifying the main drawbacks and limitations is based on the following steps: (1) analyzing evaluated methods and tools; (2) identifying which methods do not perform well in the datasets; (3) searching and analyzing the sentence pairs

**Table 8. Pearson (r) and Spearman (ρ) correlation values (0.xxx) which will be obtained in our experiments from the evaluation of all sentence similarity methods detailed below in the BIOSSES [20], MedSTS$_{full}$ [49], and CTR [50] datasets.**

| ID | Sentence similarity methods | BIOSSES | | MedSTS$_{full}$ | | CTR | |
|----|------------------------------|---------|---------|---------|---------|---------|---------|
| | | r | ρ | r | ρ | r | ρ |
| M1 | Qgram | .xxx | .xxx | .xxx | .xxx | .xxx | .xxx |
| M2 | Jaccard | .xxx | .xxx | .xxx | .xxx | .xxx | .xxx |
| M3 | Block distance | .xxx | .xxx | .xxx | .xxx | .xxx | .xxx |
| M4 | Levenshtein distance [57] | .xxx | .xxx | .xxx | .xxx | .xxx | .xxx |
| M5 | Overlap coefficient [60] | .xxx | .xxx | .xxx | .xxx | .xxx | .xxx |
| M6 | WBSM-Rada [20, 111] | .xxx | .xxx | .xxx | .xxx | .xxx | .xxx |
| M7 | WBSM-J&C [20, 112] | .xxx | .xxx | .xxx | .xxx | .xxx | .xxx |
| M8 | WBSM-cosJ&C [20, 43, 109] | .xxx | .xxx | .xxx | .xxx | .xxx | .xxx |
| M9 | WBSM-coswJ&C [20, 43, 109] | .xxx | .xxx | .xxx | .xxx | .xxx | .xxx |
| M10 | WBSM-Cai [20, 110] | .xxx | .xxx | .xxx | .xxx | .xxx | .xxx |
| M11 | UBSM-Rada [20, 111] | .xxx | .xxx | .xxx | .xxx | .xxx | .xxx |
| M12 | UBSM-J&C [20, 112] | .xxx | .xxx | .xxx | .xxx | .xxx | .xxx |
| M13 | UBSM-cosJ&C [20, 43, 109] | .xxx | .xxx | .xxx | .xxx | .xxx | .xxx |
| M14 | UBSM-coswJ&C [20, 43, 109] | .xxx | .xxx | .xxx | .xxx | .xxx | .xxx |
| M15 | UBSM-Cai [20, 110] | .xxx | .xxx | .xxx | .xxx | .xxx | .xxx |
| M16 | COM [20] | .xxx | .xxx | .xxx | .xxx | .xxx | .xxx |
| M17 | Flair [37, 77] | .xxx | .xxx | .xxx | .xxx | .xxx | .xxx |
| M18 | Pyysalo et al. [73] | .xxx | .xxx | .xxx | .xxx | .xxx | .xxx |
| M19 | BioConceptVec$_{word2vec\_sg}$ | .xxx | .xxx | .xxx | .xxx | .xxx | .xxx |
| M20 | BioConceptVec$_{word2vec\_cbow}$ | .xxx | .xxx | .xxx | .xxx | .xxx | .xxx |
| M21 | Newman-Griffis$_{word2vec\_sg}$ [70] | .xxx | .xxx | .xxx | .xxx | .xxx | .xxx |
| M22 | Newman-Griffis$_{word2vec\_cbow}$ [70] | .xxx | .xxx | .xxx | .xxx | .xxx | .xxx |
| M23 | Newman-Griffis$_{glove}$ | .xxx | .xxx | .xxx | .xxx | .xxx | .xxx |
| M24 | BioConceptVec$_{glove}$ [71] | .xxx | .xxx | .xxx | .xxx | .xxx | .xxx |
| M25 | BioWordVec$_{int}$ [22] | .xxx | .xxx | .xxx | .xxx | .xxx | .xxx |
| M26 | BioWordVec$_{ext}$ [22] | .xxx | .xxx | .xxx | .xxx | .xxx | .xxx |
| M27 | BioNLP2016$_{win2}$ [114] | .xxx | .xxx | .xxx | .xxx | .xxx | .xxx |
| M28 | BioNLP2016$_{win30}$ [114] | .xxx | .xxx | .xxx | .xxx | .xxx | .xxx |
| M29 | BioConceptVec$_{fastText}$ | .xxx | .xxx | .xxx | .xxx | .xxx | .xxx |
| M30 | USE [115] | .xxx | .xxx | .xxx | .xxx | .xxx | .xxx |
| M31 | BioSentVec (PubMed+MIMIC-III) | .xxx | .xxx | .xxx | .xxx | .xxx | .xxx |
| M32 | FastText-SkGr-BioC (this work) | .xxx | .xxx | .xxx | .xxx | .xxx | .xxx |
| M33 | BioBERT Base 1.0 (+ PubMed) | .xxx | .xxx | .xxx | .xxx | .xxx | .xxx |
| M34 | BioBERT Base 1.0 (+ PMC) | .xxx | .xxx | .xxx | .xxx | .xxx | .xxx |
| M35 | BioBERT Base 1.0 (+ PubMed + PMC) | .xxx | .xxx | .xxx | .xxx | .xxx | .xxx |
| M36 | BioBERT Base 1.1 (+ PubMed) | .xxx | .xxx | .xxx | .xxx | .xxx | .xxx |
| M37 | BioBERT Large 1.1 (+ PubMed) | .xxx | .xxx | .xxx | .xxx | .xxx | .xxx |
| M38 | NCBI-BlueBERT Base PubMed | .xxx | .xxx | .xxx | .xxx | .xxx | .xxx |
| M39 | NCBI-BlueBERT Large PubMed | .xxx | .xxx | .xxx | .xxx | .xxx | .xxx |
| M40 | NCBI-BlueBERT Base PubMed + MIMIC-III | .xxx | .xxx | .xxx | .xxx | .xxx | .xxx |
| M41 | NCBI-BlueBERT Large PubMed + MIMIC-III | .xxx | .xxx | .xxx | .xxx | .xxx | .xxx |
| M42 | SciBERT | .xxx | .xxx | .xxx | .xxx | .xxx | .xxx |
| M43 | ClinicalBERT | .xxx | .xxx | .xxx | .xxx | .xxx | .xxx |
| M44 | PubMedBERT (abstracts) | .xxx | .xxx | .xxx | .xxx | .xxx | .xxx |
| M45 | PubMedBERT (abstracts + full text) | .xxx | .xxx | .xxx | .xxx | .xxx | .xxx |
| M46 | ouBioBERT-Base, Uncased | .xxx | .xxx | .xxx | .xxx | .xxx | .xxx |

**Table 9. Pearson (r) and Spearman ($\rho$) correlation values (0.xxx) which will be obtained in our experiments from the evaluation of ontology similarity methods detailed below in the MedSTS$_{full}$ [49] dataset for each NER tool.**

| ID | Methods | MetaMap | | MetaMap Lite | | cTAKES | |
|---|---|---|---|---|---|---|---|
| | | r | $\rho$ | r | $\rho$ | r | $\rho$ |
| M11 | UBSM-Rada [20, 111] | .xxx | .xxx | .xxx | .xxx | .xxx | .xxx |
| M12 | UBSM-J&C [20, 112] | .xxx | .xxx | .xxx | .xxx | .xxx | .xxx |
| M13 | UBSM-cosJ&C [20, 43, 109] | .xxx | .xxx | .xxx | .xxx | .xxx | .xxx |
| M14 | UBSM-coswJ&C [20, 43, 109] | .xxx | .xxx | .xxx | .xxx | .xxx | .xxx |
| M15 | UBSM-Cai [20, 110] | .xxx | .xxx | .xxx | .xxx | .xxx | .xxx |
| M16 | COM [20] | .xxx | .xxx | .xxx | .xxx | .xxx | .xxx |

in which the methods report the largest differences from the gold standard; and finally, (4) analyzing and hypothesizing why the methods fail. We have already identified some of the drawbacks of several methods during our literature review and prototyping stage as follows. First, most methods reported in the literature neither consider the structure of the sentences nor the intrinsic relations between the parts that conform them. Second, BERT-based methods are trained for downstream tasks, using a supervised approach, and do not perform well in an unsupervised context. Finally, we expect to find drawbacks and limitations by analyzing and studying the results.

## Conclusions and future work

We have introduced a detailed experimental setup to reproduce, evaluate, and compare the most extensive set of methods on biomedical sentence similarity reported in the literature,

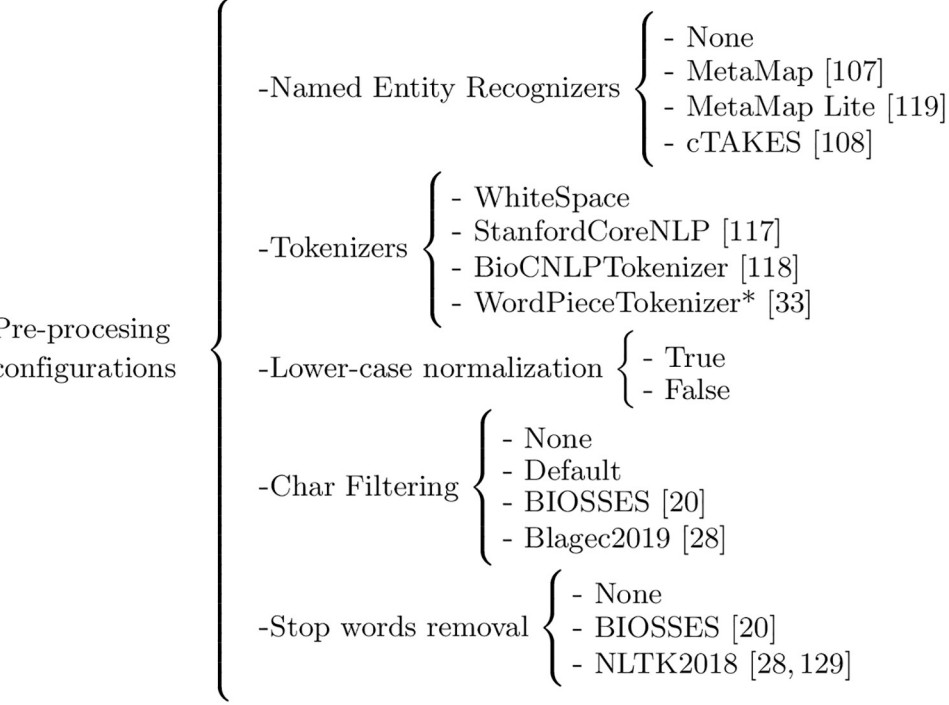

**Fig 6. Details of the pre-processing configurations that will be evaluated in this work.** (*) WordPieceTokenizer [33] will be used only for BERT-based methods. [20, 28, 33, 107, 108, 117–119, 129].

with the following aims: (1) elucidating the state of the art on the problem, (2) studying the impact of different pre-processing configurations; (3) studying the impact of the NER tools; and (4) identifying the main drawbacks and limitations of the current methods to set new lines of research. Our work also introduces the first collection of self-contained and reproducible benchmarks on biomedical sentence similarity based on the same software platform. In addition, we have proposed the evaluation of a new word embedding model based on FastText and trained on the full text of the articles in the PMC-BioC corpus [19], and the evaluation for the first time of the CTR [50] dataset.

All experiments introduced herein will be implemented into the same software library, called HESML-STS, which will be developed especially for this work. We will provide a detailed reproducibility protocol, together with a collection of software tools and a reproducibility dataset, to allow the exact replication of all our experiments, methods, and results. Thus, our reproducible experiments could be independently reproduced and extended by the research community, with the hope of becoming a de facto experimentation platform for this research line.

As forthcoming activities, we plan to evaluate the sentence similarity methods in an extrinsic task, such as semantic medical indexing [130] or summarization [131]. We also consider the evaluation of further pre-processing configurations, such as biomedical NER systems based on recent Deep Learning techniques [10], or extending our experiments and research to the multilingual scenario by integrating multilingual biomedical NER systems like Cimind [132]. Finally, we plan to evaluate some recent biomedical concept embeddings based on MeSH [133], which has not been evaluated in the sentence similarity task yet.

## Acknowledgments

We are grateful to Gizem Sogancioglu and Kathrin Blagec for answering kindly our questions to replicate their methods and experiments. UMLS CUI codes, SNOMED-CT US ontology and MeSH thesaurus were used in our experiments by courtesy of the National Library of Medicine of the United States. Finally, we are grateful to the anonymous reviewers for their valuable comments to improve the quality of the paper.

## Author Contributions

**Conceptualization:** Alicia Lara-Clares, Juan J. Lastra-Díaz, Ana Garcia-Serrano.

**Formal analysis:** Alicia Lara-Clares, Juan J. Lastra-Díaz.

**Funding acquisition:** Ana Garcia-Serrano.

**Investigation:** Alicia Lara-Clares.

**Methodology:** Alicia Lara-Clares, Juan J. Lastra-Díaz, Ana Garcia-Serrano.

**Resources:** Alicia Lara-Clares.

**Supervision:** Juan J. Lastra-Díaz, Ana Garcia-Serrano.

**Validation:** Alicia Lara-Clares.

**Visualization:** Juan J. Lastra-Díaz.

**Writing – original draft:** Alicia Lara-Clares.

**Writing – review & editing:** Juan J. Lastra-Díaz, Ana Garcia-Serrano.

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
