## [Decision Letter · Decision Letter 0]

24 Dec 2020

PONE-D-20-35183

A reproducible experimental survey on biomedical sentence similarity

PLOS ONE

Dear Dr. Lara Clares,

Thank you for submitting your manuscript to PLOS ONE. After careful consideration, we feel that it has merit but does not fully meet PLOS ONE’s publication criteria as it currently stands. Therefore, we invite you to submit a revised version of the manuscript that addresses the points raised during the review process.

We look forward to receiving your revised manuscript.

Kind regards,

Bridget McInnes, Ph.D.

Academic Editor

PLOS ONE

Journal Requirements:

"This work has been supported by the UNED predoctoral grant started in April 2019 372

(BICI N7, November 19th, 2018).".

i) We note that you have provided funding information that is not currently declared in your Funding Statement. However, funding information should not appear in the Acknowledgments section or other areas of your manuscript. We will only publish funding information present in the Funding Statement section of the online submission form.

ii) Please remove any funding-related text from the manuscript and let us know how you would like to update your Funding Statement. Currently, your Funding Statement reads as follows:

"UNED predoctoral grant started in April 2019 (BICI N7, November 19th, 2018)

https://www.uned.es/

The funders had and will not have a role in study design, data collection and analysis, decision to publish, or preparation of the manuscript."

 iii) Please include your amended statements within your cover letter; we will change the online submission form on your behalf.

Reviewers' comments:

Reviewer's Responses to Questions

**Comments to the Author**

1. Does the manuscript provide a valid rationale for the proposed study, with clearly identified and justified research questions?

Reviewer #1: Yes

Reviewer #2: Partly

2. Is the protocol technically sound and planned in a manner that will lead to a meaningful outcome and allow testing the stated hypotheses?

Reviewer #1: Partly

Reviewer #2: Partly

3. Is the methodology feasible and described in sufficient detail to allow the work to be replicable?

Reviewer #1: Yes

Reviewer #2: No

4. Have the authors described where all data underlying the findings will be made available when the study is complete?

Reviewer #1: Yes

Reviewer #2: Yes

5. Is the manuscript presented in an intelligible fashion and written in standard English?

Reviewer #1: Yes

Reviewer #2: No

6. Review Comments to the Author

You may also provide optional suggestions and comments to authors that they might find helpful in planning their study.

Reviewer #1: This well-written paper describes a project that is of substantial importance to the domain. I have just a few questions and suggestions:

1. What is the difference between RQ1 and RQ2? Please provide more detail in these questions.

2. Please change line 185 from "and named entities, that they have in common or not; and finally. And finally,"

to

"and named entities, that they have in common or not. Finally,"

3. Please add information on how you will handle unexpected barriers while conducting experiments. In the spirit of this requirement from the publisher - "As there may be aspects of the methodology and analysis which can only be refined once the work is undertaken, authors should outline potential assumptions and explicitly describe what aspects of the proposed analyses, if any, are exploratory." - please briefly discuss contingency efforts to handle the unexpected.

4. It would be so useful if you could do a small pilot study, using your system, and report the results.

Reviewer #2: The authors detail in this registered report the design and a project of implementation of a large and reproducible experimental survey on sentence similarity measures for the biomedical domain.

A first concern is related to the research questions, which may result in two or three (RQ1 to RQ5 are similar).

I would suggest the authors to cut this part and to better highlight the motivations instead.

Several sentences are repeated more than twice (e.g. lines 111 to 114, and lines 117 to 120)

The MeSH thesaurus is not an ontology !!

The authors first introduce a categorization of the literature on sentence semantic similarity measures for the general and biomedical language domains, mainly founded on already research. They identify which methods that will be reproduced and evaluated in "planned experiments". How was conducted the choice/selection of those methods ? It is not obvious.

What was the strategy to identify the related publications ? Was it inspired from systematic review or meta-analysis ?

With more than 100 references, the survey is very well documented. I would suggest the authors to have also a look to the following survey on clinical natural processing : "Hahn U, Oleynik M. Medical Information Extraction in the Age of Deep Learning. Yearb Med Inform. 2020 Aug;29(1):208-220. doi: 10.1055/s-0040-1702001. Epub 2020 Aug 21. PMID: 32823318; PMCID: PMC7442512."

The authors detail their experimental setup by describing their evaluation's plan in order to compare most of the sentence similarity methods with seven goals related to the research questions and motivation.

The first step of the pipeline is preprocessing with four stages. Nothing is said here about the choices. Why UMLS ? Why MetaMap and what are the features/options ? why teokenizing ? what is the impact of tokenizing instead of using lexicons ? What is the original aim of preprocessing ?

I would suggest the authors to detail give the ref of M8, M9, M10, M13, M14, ... (line 265) instead of letting the readers to refer to table 2 or table 3...

line 303 MedSTS or MedSTS full ? what is the difference ?

In don't think that the table 7 (planned results...) will be clear enough to help in answering the research questions enumerated in page 3. Moreover, the registered report lacks a section in which the authors would explain how it is intended to answer to the RQ, which step of the pipeline/workflow is concerned to the RQ. It also lacks a gantt with the tasks and the schedule/timeline. Since a large part of the work still needs to be developed, what kind of methodological software development is envisaged ? What are the technological issues/locks to be removed ?

Finally, what would be the general conclusion of the comparison? A set of proposed methods ? To do what ? Since the application domains of sentence similarity, categorizing the methods according to the aim of sentence similarity is another axis to be considered here.

Minor comments :

line 34 desambiguate acronym MeSH

line 37 desambiguate MIMIC

line 39 Pubmed -> PubMed

line 42 desambiguate BERT

line 64 desambiguate HESML

line 85 desambigiate NER

line 105 BIOSSES ?

line 106 MedSTS, CTR ?

line 164 SNOMED-CT ?

line 173 SWEM ?

line 185 "; and finally. And finally," ?

line 192 WBSM? UBSM?

line 212 ELMo ?

line 226 Metamap  MetaMap

line 233 UMLS ?

line 234 METAMAP  MetaMap

line 237 char ?

Incomplete references

43. Lastra-D ´ıaz JJ, Goikoetxea J, Hadj Taieb MA, Garc ´ıa-Serrano A, Aouicha MB, Agirre E. A reproducible survey on word embeddings and ontology-based methods for word similarity: linear combinations outperform the state of the art. Engineering Applications of Artificial Intelligence. 2019;.

57 Agirre E, Banea C, Cer D, Diab M, others. Semeval-2016 task 1: Semantic textual similarity, monolingual and cross-lingual evaluation. Workshop on Semantic . . . . 2016;.

58. Cer D, Diab M, Agirre E, Lopez-Gazpio I, Specia L. SemEval-2017 Task 1: Semantic Textual Similarity - Multilingual and Cross-lingual Focused Evaluation. arXiv. 2017;.

69. Shen D, Wang G, Wang W, Min MR, Su Q, Zhang Y, et al. Baseline Needs More Love: On Simple Word-Embedding-Based Models and Associated Pooling Mechanisms. arXiv. 2018;.

71. Chen Q, Du J, Kim S, Wilbur WJ, Lu Z. Combining rich features and deep learning for finding similar sentences in electronic medical records. Proc of the BioCreative/OHNLP Challenge. 2018;.

81. Pawar A, Mago V. Calculating the similarity between words and sentences using a lexical database and corpus statistics. arXiv. 2018;.

86. Newman-Griffis D, Lai AM, Fosler-Lussier E. Insights into Analogy Completion from the Biomedical Domain. arXiv. 2017;.

91. Arora S, Liang Y, Ma T. A Simple but Tough-to-Beat Baseline for Sentence Embeddings; 2016.

95. Beltagy I, Lo K, Cohan A. SciBERT: A Pretrained Language Model for Scientific Text. arXiv. 2019;.

96. Gu Y, Tinn R, Cheng H, Lucas M, Usuyama N, Liu X, et al. Domain-Specific Language Model Pretraining for Biomedical Natural Language Processing. arXiv. 2020;.

97. Wada S, Takeda T, Manabe S, Konishi S, Kamohara J, Matsumura Y. A pre-training technique to localize medical BERT and enhance BioBERT. arXiv. 2020;.

100. Al-Natsheh HT, Martinet L, Muhlenbach F, others. Udl at semeval-2017 task 1: Semantic textual similarity estimation of english sentence pairs using regression model over pairwise features. Proc of the 11th semeval conference. 2017;.

116. Cer D, Yang Y, Kong SY, Hua N, Limtiaco N, St John R, et al. Universal Sentence Encoder. arXiv. 2018;.

117. Huang K, Altosaar J, Ranganath R. ClinicalBERT: Modeling Clinical Notes and Predicting Hospital Readmission. arXiv. 2019;.

72. P Jaccard.  Jaccard P

7. PLOS authors have the option to publish the peer review history of their article (what does this mean?). If published, this will include your full peer review and any attached files.

Reviewer #1: No

Reviewer #2: **Yes: **Lina F SOUALMIA

---

## [Author Response · Author response to Decision Letter 0]

25 Jan 2021

Dear editor and reviewers,

We are very grateful for your significant effort to review our manuscript, as well as your kind remarks and suggestions to improve the quality of the paper.

We have accepted and followed most suggestions made by the reviewers, except for the recommendation of providing pilot data, which we have discarded after making a series of prototypes to eliminate all risks, both in development and publication, as we have detailed in the new section "Software integration and contingency plan".

We have also modified the manuscript's title to “Protocol for a reproducible experimental survey on biomedical sentence similarity”, to make it more consistent with the work we are presenting. Thus, if this submission is finally accepted, our next work implementing the proposed protocol would be entitled “A reproducible experimental survey on biomedical sentence similarity”. In addition, we have fixed a minor error in Figure 1 by splitting Word2vec-based methods into two categories: Skip-gram and CBOW methods. 

Finally, we provide below our detailed answer for each suggestion made by the reviewers.

Most sincerely,

The authors

Journal Requirements:

[Authors] We have removed the Postal Codes and street addresses from the author's affiliation.

[Authors] We will publish all the necessary software tools, source code, data, and a detailed reproducibility protocol to allow the exact replication of all the experiments and results proposed in this paper. In addition, the aforementioned software tools, data, and reproducibility documentation will be introduced in another companion data article entitled “Reproducibility protocol and dataset for a large reproducible survey on biomedical sentence similarity”, which will be co-submitted with our next paper implementing the experimental protocol introduced in this work. Therefore, we cannot provide the DOIs in this paper because all the software tools and data have not been developed yet, and they will be a significant product of our research.

"This work has been supported by the UNED predoctoral grant started in April 2019 372

(BICI N7, November 19th, 2018).".

i) We note that you have provided funding information that is not currently declared in your Funding Statement. However, funding information should not appear in the Acknowledgments section or other areas of your manuscript. We will only publish funding information present in the Funding Statement section of the online submission form.

ii) Please remove any funding-related text from the manuscript and let us know how you would like to update your Funding Statement. Currently, your Funding Statement reads as follows:

"UNED predoctoral grant started in April 2019 (BICI N7, November 19th, 2018)

https://www.uned.es/

The funders had and will not have a role in study design, data collection and analysis, decision to publish, or preparation of the manuscript."

 iii) Please include your amended statements within your cover letter; we will change the online submission form on your behalf.

 [Authors] We have removed the funding information from the Acknowledgments section, thank you for the clarification. The correct funding information is as follows:

“UNED predoctoral grant started in April 2019 (BICI N7, November 19th, 2018)

https://www.uned.es/

The funders had and will not have a role in study design, data collection and analysis, decision to publish, or preparation of the manuscript.”

Reviewer #1: 

This well-written paper describes a project that is of substantial importance to the domain. I have just a few questions and suggestions.

[Authors] Thank you very much for your effort.

What is the difference between RQ1 and RQ2? Please provide more detail in these questions.

[Authors] Done. Following your suggestion and another similar one made by the second reviewer, we have grouped and simplified our research questions for the sake of clarity. We have substituted RQ1 by RQ2, and we have reduced the specific research questions to five (see lines 81-89 on page 3).

Please change line 185 from "and named entities, that they have in common or not; and finally. And finally," to "and named entities, that they have in common or not. Finally,"

[Authors] Done (see line 204-205 on page 6). 

Please add information on how you will handle unexpected barriers while conducting experiments. In the spirit of this requirement from the publisher - "As there may be aspects of the methodology and analysis which can only be refined once the work is undertaken, authors should outline potential assumptions and explicitly describe what aspects of the proposed analyses, if any, are exploratory." - please briefly discuss contingency efforts to handle the unexpected.

[Authors] We have included a subsection called “Software integration and contingency plan” detailing the actions carried out to mitigate the impact of risks and unexpected problems (see the section starting at line 332-381 on pages 10-13).

It would be so useful if you could do a small pilot study, using your system, and report the results.

[Authors] DISMISSED. Following your previous suggestion, we have included a contingency plan (see remark above). We have also carried out the necessary actions to minimize the risks in the development of our study. 

Reviewer #2:

A first concern is related to the research questions, which may result in two or three (RQ1 to RQ5 are similar). I would suggest the authors to cut this part and to better highlight the motivations instead. 

[Authors] Thank you very much for your effort.

[Authors] Done (see lines 81-89 on page 3). Following your suggestion, and other similar made by the first reviewer, we have removed RQ1, RQ4, and RQ5 for the sake of clarity.

Several sentences are repeated more than twice (e.g. lines 111 to 114, and lines 117 to 120) 

[Authors] Done (see lines 119-121 on page 4).

The MeSH thesaurus is not an ontology !!

[Authors] Done. Thank you for the clarification (see line 186 on page 5, and Figure 5).

The authors first introduce a categorization of the literature on sentence semantic similarity measures for the general and biomedical language domains, mainly founded on already research. They identify which methods that will be reproduced and evaluated in "planned experiments". How was conducted the choice/selection of those methods ? It is not obvious. What was the strategy to identify the related publications ? Was it inspired from systematic review or meta-analysis ?

[Authors] We have included a paragraph entitled “Literature review methodology” to explain the methodology used to carry out our review of the literature, and the selection of methods which will be reproduced (see lines 145-161 on pages 4-5). Likewise, we have included a paragraph in Section “Selection of methods” detailing the methodology for selecting the methods that will be evaluated in this work (see lines 250-254 on page 8).

With more than 100 references, the survey is very well documented. I would suggest the authors to have also a look to the following survey on clinical natural processing : "Hahn U, Oleynik M. Medical Information Extraction in the Age of Deep Learning. Yearb Med Inform. 2020 Aug;29(1):208-220. doi: 10.1055/s-0040-1702001. Epub 2020 Aug 21. PMID: 32823318; PMCID: PMC7442512."

[Authors] We have cited the publication mentioned above (see line 7 on page 1). We have also considered evaluating more NER tools as future work (see lines 559-561 on page 22).

The authors detail their experimental setup by describing their evaluation's plan in order to compare most of the sentence similarity methods with seven goals related to the research questions and motivation. The first step of the pipeline is preprocessing with four stages. Nothing is said here about the choices. Why UMLS ? Why MetaMap and what are the features/options ? why tokenizing ? what is the impact of tokenizing instead of using lexicons ? What is the original aim of preprocessing ?

[Authors] In order to explain the selection criteria of the pre-processing components for our experiments, we have carried out the following revisions. First, we have moved the subsection “Selection of methods” above the subsection “Detailed workflow of our experiments”.

[Authors] And second, we have added a new subsection “Selection of language processing methods and tools”, which answers the five questions above (see lines 289-331 on pages 8-10).

I would suggest the authors to detail give the ref of M8, M9, M10, M13, M14, ... (line 265) instead of letting the readers to refer to table 2 or table 3… 

[Authors] Done (see lines 265-267 on page 8, and line 285 on page 8).

line 303 MedSTS or MedSTS full ? what is the difference ?

[Authors] Done, we have included a proper explanation in the first paragraph of section “Methods proposed for the general language domain” (see lines 173-175 on page 5). 

I don't think that the table 7 (planned results...) will be clear enough to help in answering the research questions enumerated in page 3. Moreover, the registered report lacks a section in which the authors would explain how it is intended to answer to the RQ, which step of the pipeline/workflow is concerned to the RQ. It also lacks a gantt with the tasks and the schedule/timeline. Since a large part of the work still needs to be developed, what kind of methodological software development is envisaged ? What are the technological issues/locks to be removed ?

[Authors] First, in order to clarify how is intended to answer to the RQ, we have included a new Section called “Answering our research questions” (see lines 490-536 on page 20), in which we detail how to use our data to answer the RQs. 

[Authors] Second, as you suggest, in order to fix the lack of a timeline, we have renamed the section “Software implementation” to “Software implementation and development plan”, and we have included a detailed software development plan (see line 461-472 on page 16). 

[Authors] Third, as regards your questions related to the development methodology and the potential risks, we have included a section called “Software integration and contingency plan” (see lines 332-381 on pages 10-13) to answer both questions. This new section explains how we have identified and mitigated potential risks, a question which was also pointed out by the first reviewer.

Finally, what would be the general conclusion of the comparison? A set of proposed methods ? To do what ? Since the application domains of sentence similarity, categorizing the methods according to the aim of sentence similarity is another axis to be considered here.

[Authors] The main aim of this work is to determine which are the best performing methods in this line of research in a sound, reproducible, and conclusively manner by comparing them to a common software platform. A second aim is to study the impact of the pre-processing stage and the NER tools on the performance of the methods under study. And finally, the final aim, but not less important, is to elucidate the main drawbacks and limitations of the current methods to identify new research lines. We have modified the “Conclusions and future work” section to clarify the expected conclusions of this work (see lines 539-545 and lines 553-563 on pages 21-22).

Minor errors: 

line 34 desambiguate acronym MeSH Done (see line 34 on page 2).

line 37 desambiguate MIMIC Done (see line 37 on page 2).

line 39 Pubmed -> PubMed Done throughout (see line 40 on page 2, and Tables 3 and 4).

line 42 desambiguate BERT Done (see line 44 on page 2).

line 64 desambiguate HESML Done (see line 67 on page 3).

line 85 desambigiate NER Done (see line 84 on page 3).

line 105 BIOSSES ? Done (see line 106 on page 4).

line 106 MedSTS, CTR ? Done (see lines 107-109 on page 4).

line 164 SNOMED-CT ? Done (see lines 184 on page 5).

line 173 SWEM ? Done (see line 192 on page 5).

line 185 "; and finally. And finally," ? Done, we have modified the sentences as suggested by the first reviewer (see lines 204 on page 6).

line 192 WBSM? UBSM? Done (see lines 211-212 on page 6).

line 212 ELMo ? Done (see line 231 on page 6).

line 226 Metamap  MetaMap Done (see line 246 on page 6).

line 233 UMLS ? Done (see line 321 on page 9).

line 234 METAMAP  MetaMap Done (see line 388 on page 13).

line 237 char ? Done (see line 391 on page 13).

Incomplete references:

General comment from the authors: We have replaced most Arxiv references as possible.

43. Lastra-D ´ıaz JJ, Goikoetxea J, Hadj Taieb MA, Garc ´ıa-Serrano A, Aouicha MB, Agirre E. A reproducible survey on word embeddings and ontology-based methods for word similarity: linear combinations outperform the state of the art. Engineering Applications of Artificial Intelligence. 2019;. 

Done.

57 Agirre E, Banea C, Cer D, Diab M, others. Semeval-2016 task 1: Semantic textual similarity, monolingual and cross-lingual evaluation. Workshop on Semantic . . . . 2016;. Done.

58. Cer D, Diab M, Agirre E, Lopez-Gazpio I, Specia L. SemEval-2017 Task 1: Semantic Textual Similarity - Multilingual and Cross-lingual Focused Evaluation. arXiv. 2017;.

Done.

69. Shen D, Wang G, Wang W, Min MR, Su Q, Zhang Y, et al. Baseline Needs More Love: On Simple Word-Embedding-Based Models and Associated Pooling Mechanisms. arXiv. 2018;.

Done.

71. Chen Q, Du J, Kim S, Wilbur WJ, Lu Z. Combining rich features and deep learning for finding similar sentences in electronic medical records. Proc of the BioCreative/OHNLP Challenge. 2018;.

Done.

81. Pawar A, Mago V. Calculating the similarity between words and sentences using a lexical database and corpus statistics. arXiv. 2018;.

Done.

86. Newman-Griffis D, Lai AM, Fosler-Lussier E. Insights into Analogy Completion from the Biomedical Domain. arXiv. 2017;.

Done.

91. Arora S, Liang Y, Ma T. A Simple but Tough-to-Beat Baseline for Sentence Embeddings; 2016.

Done.

95. Beltagy I, Lo K, Cohan A. SciBERT: A Pretrained Language Model for Scientific Text. arXiv. 2019;.

Done.

96. Gu Y, Tinn R, Cheng H, Lucas M, Usuyama N, Liu X, et al. Domain-Specific Language Model Pretraining for Biomedical Natural Language Processing. arXiv. 2020;.

Done 

97. Wada S, Takeda T, Manabe S, Konishi S, Kamohara J, Matsumura Y. A pre-training technique to localize medical BERT and enhance BioBERT. arXiv. 2020;.

Done

100. Al-Natsheh HT, Martinet L, Muhlenbach F, others. Udl at semeval-2017 task 1: Semantic textual similarity estimation of english sentence pairs using regression model over pairwise features. Proc of the 11th semeval conference. 2017;.

Done

116. Cer D, Yang Y, Kong SY, Hua N, Limtiaco N, St John R, et al. Universal Sentence Encoder. arXiv. 2018;.

Done

117. Huang K, Altosaar J, Ranganath R. ClinicalBERT: Modeling Clinical Notes and Predicting Hospital Readmission. arXiv. 2019;.

Done.

P Jaccard.  Jaccard P 

[Authors] Done (see reference 74)

---

## [Decision Letter · Decision Letter 1]

3 Mar 2021

Protocol for a reproducible experimental survey on biomedical sentence similarity

PONE-D-20-35183R1

Dear Dr. Lara Clares,

We’re pleased to inform you that your manuscript has been judged scientifically suitable for publication and will be formally accepted for publication once it meets all outstanding technical requirements.

Kind regards,

Bridget McInnes, Ph.D.

Academic Editor

PLOS ONE

Additional Editor Comments (optional):

Reviewers' comments:

Reviewer's Responses to Questions

**Comments to the Author**

1. Does the manuscript provide a valid rationale for the proposed study, with clearly identified and justified research questions?

Reviewer #1: Yes

Reviewer #2: Yes

2. Is the protocol technically sound and planned in a manner that will lead to a meaningful outcome and allow testing the stated hypotheses?

Reviewer #1: Yes

Reviewer #2: Yes

3. Is the methodology feasible and described in sufficient detail to allow the work to be replicable?

Reviewer #1: Yes

Reviewer #2: Yes

4. Have the authors described where all data underlying the findings will be made available when the study is complete?

Reviewer #1: Yes

Reviewer #2: Yes

5. Is the manuscript presented in an intelligible fashion and written in standard English?

Reviewer #1: Yes

Reviewer #2: Yes

6. Review Comments to the Author

You may also provide optional suggestions and comments to authors that they might find helpful in planning their study.

Reviewer #1: Thank you for addressing my concerns.

The manuscript provide a valid rationale for the proposed study, with clearly identified and justified research questions.

The methodology is feasible and described in detail.

The authors have addressed data access issues for the project as a whole.

The manuscript is well-written, with useful graphics.

Reviewer #2: The authors have significantly improved their manuscript. A lot of work have been done and all the comments of the reviewers have been taken into consideration, including re-writing of several sections, in this revised version.

7. PLOS authors have the option to publish the peer review history of their article (what does this mean?). If published, this will include your full peer review and any attached files.

Reviewer #1: No

Reviewer #2: **Yes: **Lina F. Soualmia

---

## [Editor Report · Acceptance letter]

8 Mar 2021

PONE-D-20-35183R1 

Protocol for a reproducible experimental survey on biomedical sentence similarity 

Dear Dr. Lara-Clares:

I'm pleased to inform you that your manuscript has been deemed suitable for publication in PLOS ONE. Congratulations! Your manuscript is now with our production department. 

Kind regards, 

on behalf of

Dr. Bridget McInnes 

Academic Editor

PLOS ONE